# Method for Recognizing Pressing Position and Shear Force Using Active Acoustic Sensing on Gel Plates

**DOI:** 10.3390/s22249951

**Published:** 2022-12-16

**Authors:** Hiroki Watanabe, Kaito Sasaki, Tsutomu Terada, Masahiko Tsukamoto

**Affiliations:** 1Graduate School of Information Science and Technology, Hokkaido University, Sapporo 060-0814, Japan; 2Graduate School of Engineering, Kobe University, Kobe 657-8501, Japan

**Keywords:** touch interface, pressing position recognition, shear-force recognition, active acoustic sensing

## Abstract

A touch interface is an important technology used in many devices, including touch panels in smartphones. Many touch panels only detect the contact position. If devices can detect shear force in addition to the contact position, various touch interactions are possible. We propose a two-step recognition method for recognizing the pressing position and shear force using active acoustic sensing, which transmits acoustic signals to an object and recognizes the state of the object by analyzing its response. Specifically, we attach a contact speaker transmitting an ultrasonic sweep signal and a contact microphone receiving ultrasonic waves to a plate of gel. The propagation characteristics of ultrasonic waves differ due to changes in the shape of the gel caused by the user’s actions on the gel. This system recognizes the pressing position and shear force on the basis of the difference in acoustic characteristics. An evaluation of our method involving a user-independent model confirmed that four pressing positions were recognized with an F1 score of 85.4%, and four shear-force directions were recognized with an F1 score of 69.4%.

## 1. Introduction

A touch interface is an important technology used in many devices, e.g., smartphones, tablets, and smartwatches. There are several detection methods for touch panels [1]; however, many only detect the contact position. In addition to position detection, pressure, and shear force, which pushes tangentially, sensing capabilities enable various touch interactions. In terms of pressure sensing, Apple’s iPhone series released in 2015 added the ability to detect vertical pressure with a pressure-sensitive sensor that enables shortcuts in operation on the basis of the strength of the push [2]. However, this function is unavailable on the current iPhone series, and few other touch panels have similar functions. One possible reason for this is that different sensors are required for position detection and pressure detection, leading to increased costs.

Most computer interfaces are composed of hard materials. However, interfaces using soft materials have been proposed [3]. Nakai et al. proposed a method for estimating shear force from the force applied to an elastic material between a sheet overlaid on the touch panel and enclosure [4]. If the direction of the shear force is available, we can add a new input method that is distinguishable from conventional swipes. However, the detection of the pressing position is based on the touch panel, and the number of input patterns that can be identified with a single sensor is limited relative to the deforming characteristics of the material. It is generally necessary to embed a hard sensor into a soft material or to use multiple detection mechanisms to detect multiple deformation patterns with a soft sensor, which increases the cost and complexity of the system. Therefore, it is useful to detect multiple operations with a soft sensor that consists of fewer detection mechanisms.

We propose a two-step recognition method for recognizing pressing position and shear force using active acoustic sensing that transmits acoustic signals to an object and recognizes the state of the object on the basis of the response. Specifically, we attached a contact speaker transmitting an ultrasonic sweep signal and a contact microphone receiving ultrasonic waves onto a plate of gel. The propagation characteristics of ultrasonic waves differ due to changes in the shape of the gel caused by the user’s actions on the gel. This system recognizes the pressing position and shear force on the basis of the difference in acoustic characteristics. The advantage of the proposed method is that it can simultaneously recognize the pressing position and shear force with a simple system configuration consisting of a contact speaker and contact microphone. The two-step recognition function of our method recognizes the direction of shear force after recognizing the pressing position.

We evaluated the proposed method on four pressing positions and for shear-force directions. Evaluation results indicate that position recognition was 89.2% for a naive method with one-step recognition and 89.4% for the proposed two-step recognition method using a user-independent model. For the shear-force direction, the one-step recognition method resulted in an F1 score of 65.5% and the proposed method resulted in an F1 score of 69.4%, confirming the effectiveness of the proposed method.

## 2. Related Work

### 2.1. Touch-Sensing Technology

Capacitive touch sensors used in smartphones can achieve small form factors. However, it can only measure the area of contact, and cannot reliably discriminate between different pressure levels because it is based on capacitance. Therefore, much research has been conducted to enrich the variety of touch interactions on surfaces such as touchscreens [5,6]. PreSenseII is a device that recognizes the position, touch, and pressure of a user’s finger [7]. Resenberg et al. developed the multitouch touchpad UnMousePad [8] by using a matrix of force-variable resistors to enable flexible and inexpensive pressure-sensitive multitouch input. Quinn proposed a method for estimating the contact force on a touch panel from the smartphone’s built-in barometric pressure sensor [9]. Although these studies could detect pressure with touch panels, they could not detect shear force.

To detect shear force, interfaces using soft materials were studied [3,10]. WrinkleSurface is a system that enables the user to use actions such as pressing a finger hard against the input surface, shifting a finger, or twisting a finger for input by reading the wrinkle shape of the gel sheet attached to the touch panel with an infrared camera [11]. However, since a camera is used, it requires thickness to gain optical distance, hindering implementation in a small device. Nakai et al. proposed a method of detecting shear-force input by installing a sheet on a touch panel screen, and elastic material between the sheet and frame of the enclosure [4]. Shear force is estimated using Hooke’s law from the spring constant of the elastic material, and shear-force input is possible in addition to the conventional swipe input. Huang et al. proposed a speed-control method by overlaying a transparent thin sheet on the touchscreen of a smartphone and fixing its four corners to the corners of the smartphone with rubber bands [12]. Wei et al. proposed an all-optical tactile sensing platform that could respond to tiny shear forces, such as fingertip slipping with a low power [13]. Zhang et al. proposed a self-powered multidirectional force sensor based on triboelectric nanogenerators with a three-dimensional structure that could sense normal and shear forces in real time [14]. Zhou et al. developed a multiaxial tactile sensor based on a soft anisotropic waveguide that could distinguish between normal and shear forces [15]. However, these studies used a different detection mechanism to identify the shear force from the detection of the pressing position on the touch panel or needed specialized materials to identify the shear force. Our study is different in terms of recognizing pressing position and shear force using only a gel plate, and a contact microphone and contact speaker.

### 2.2. Active Acoustic Sensing

Active acoustic sensing is a widely used sensing technique that transmits sound waves to an object and captures the reflected sound from the object. By analyzing the captured sound, a system can recognize the state of the object [16,17,18]. Active acoustic sensing is also used for touch-sensing methods [19,20]. Ono et al. proposed a method of recognizing touch input and adding interactivity to objects by attaching a contact microphone and contact speaker to the object [21]. Iwase et al. proposed a method of identifying the type and position of objects placed on an acrylic plate by attaching microphones and a speaker to the plate [22]. Acoustruments is a system that extends interaction by using active acoustic sensing with a tubelike attachment that runs from the speaker to the microphone of a smartphone [23]. Ono et al. proposed a method of recognizing the presence or absence of contact with an object in addition to pressing pressure by attaching a contact microphone and contact speaker to the object [24].

In these studies, active acoustic sensing was used as a method of extending touch interaction by recognizing how to touch an object and the strength of the touching force. The idea of our study is similar in terms of using active acoustic sensing on an object to recognize user actions. However, the proposed method recognizes pressing positions as well as the direction of shear force using a gel plate, the acoustic properties of which change with user inputs.

## 3. Proposed Method

We assumed a system in which a contact speaker and contact microphone were attached to a gel plate, and an ultrasonic sweep signal was repeatedly transmitted from the contact speaker, as shown in Figure 1. Since the active acoustic sensing using audible sound can cause not only uncomfortable noise to the user and the surrounding people, but also misrecognition due to environmental noise, we selected an ultrasonic sweep signal (above 20 kHz). The contact microphone captures ultrasonic waves propagating through the gel, and the system calculates the fast Fourier transform (FFT). The sampling frequency used in this study was 96 kHz to capture signals up to around 40 kHz, and the number of FFT samples was 8192. As shown in Figure 2, the obtained frequency response differs with the change in the shape of the gel due to the pressing position and direction of shear force. The system can recognize the user inputs by using this difference. Since the sound emitted from the speaker is ultrasound, humans cannot hear the emitted sound.

### 3.1. Sweep Signal

We used the ultrasonic sweep signal as an acoustic signal from the contact speaker. A sweep signal has a frequency that increases/decreases with time. Since it includes various frequencies, we could acquire more features than only a fixed frequency.

The frequency of the sweep signal is expressed by the following equation:(1)$$\begin{array}{c} f\left(t\right)=\frac{f_{1}-f_{0}}{T}t+f_{0}, \end{array}$$
where *t* is time, f(t) is the frequency at *t*, $f_{0}$ is the start frequency, $f_{1}$ is the stop frequency, and *T* is the duration of the sweep signal. Therefore, the sweep signal is calculated as follows:(2)$$\begin{array}{c} \sin\left\{2\pi t\left(\frac{f_{1}-f_{0}}{2T}t+f_{0}\right)\right\}. \end{array}$$

On the basis of the literature [16,21], $f_{0}$, $f_{1}$, and *T* were set to 20 kHz, 40 kHz, and 0.02 s, respectively.

### 3.2. Recognition Method

When conducting recognition using sensor data, the obtained values are not used as they are, but feature extraction is carried out to efficiently understand the behavior. We used the linear-frequency cepstral coefficient (LFCC) as the feature value [25,26], which is a linear version of the mel-frequency cepstral coefficient (MFCC). In contrast to MFCCs, which are used for audio and speech recognition, LFCCs use a linear filter bank to reduce dimensions. We used LFCCs to equally extract features from the frequency spectrum. We used 20 filter banks and removed the first LFCC, which is the direct current component. Thus, we acquired 19 features.

Although we did not limit the classifier algorithm, we used a support vector machine as the classifier in this study.

### 3.3. Two-Step Recognition

The proposed method recognizes both pressing position and shear force. Hence, there are multiple shear-force directions at each pressing position that may result in a large number of classification classes with a single classifier. If there are *n* pressing positions and *m* shear-force directions, the classifier recognizes n×m classes, and the recognition performance is expected to deteriorate. Previous research confirmed that position recognition on acrylic plates using active acoustic sensing was achieved with high accuracy [22]; thus, we prevented the decrease in recognition rate by recognizing the pressing position first. As shown in Figure 3, in the first step, *n* classes are classified as the position recognition, and in the second step, *m* classes of shear forces corresponding to each position are classified. By first carrying out position recognition, which is expected to have high recognition accuracy, two-step recognition can improve recognition performance.

As described above, touch position recognition using active acoustic sensing was studied in a previous study [22]. The contribution of this study is that we propose simultaneous recognition of touch position and sear-force direction using active acoustic sensing and two-step recognition.

## 4. Implementation

### 4.1. Hardware

Figure 4 shows the implemented device. The gel material was a styrene-based elastomer with a thickness of 5 mm. As a physical property, it has viscoelasticity and can be fixed to a surface of an object without adhesive. Viscoelasticity can also be used for push and shear force input. Deformation caused by the push/shear force input returns to the original state when the finger is released thanks to the property of the gel. As a chemical property, the used gel is stable and does not change at room temperature or change its properties when sound waves are applied. The contact speaker and contact microphone were fixed by sandwiching them diagonally between two 10 cm square gels. This prevents changes in frequency response caused by the misalignment of the speaker and microphone. The ultrasonic sweep signal transmitted from the PC is amplified by an amplifier and then transmitted to the gel through a contact speaker. A Fostex PC200USB-HR amplifier and a Thrive OMR20F10-BP310 contact speaker (Figure 5a) were used. The contact microphone captured the ultrasonic sweep signals propagating through the gel, and the PC recorded the signals through the audio interface. The contact microphone was a Murata 7BB-20-6L0 (Figure 5b), and the audio interface was Native Instruments Komplete Audio 6. The sampling frequency for transmitting/recording signal was 96 kHz, and the number of quantization bits was 16. The PC used for transmitting the ultrasonic sweep signal, recording, and analyzing the acquired data was an Apple MacBook Pro (CPU: Intel Core i7 2.7 GHz, RAM: 16 GB).

### 4.2. Software

We used Audacity to generate the ultrasonic sweep signals and Ocenaudio to record the sound from the microphones. The software for data analysis was implemented using Python. The number of samples for the FFT was 8192. On the basis of the sampling theorem, the first half of 4096 data points were obtained. The LFCC was extracted by taking the points corresponding from 20 k to 40 kHz.

## 5. Evaluation

### 5.1. Experimental Setup

We evaluated the performance of the proposed method involving participants (seven males in their 20s). This evaluation was approved by the human ethics committee of the Graduate School of Engineering, Kobe University (Permission Number: 03-26). As shown in Figure 6, the pressing positions were P1 to P4 on the gel plate, marked with black dots. These points were 1.77 cm from the center of the gel plate in the direction of each of the four corners of the plate. There were four shear-force directions indicated by a–d in Figure 6. The data on the strength of multiple shear forces were obtained by varying the vertical pressing force and horizontal shear-force strengths. The combinations of vertical pressing force and shear force are shown in Table 1, where the shear force of 300 g was set assuming a weak input, and 600 g assuming a strong input. A six-axis force sensor (Leptrino SFS100YA500U6) was used to measure the shear force, and the participants acquired data by referring to the value of the force sensor. All participants used their index fingers for pushing the gel. The gel was pressed for approximately five seconds per operation, and 50 LFCCs were extracted. This series of actions at all pressing positions, shear-force directions, and shear-force strengths were considered to be one set, and all participants conducted two sets. Participant A was left-handed, and the rest were right-handed. As a result, we collected 44,800 data (50 LFCCs × 4 positions × 4 directions × 4 force combinations × 2 sets × 7 participants).

With the assumed system, it is desirable to use the proposed method without learning for personalization. Therefore, we used leave-one-participant-out cross-validation as an evaluation method; that is, the machine learning model was trained on data other than those of the test participants. We used the F1 score as the evaluation index, which is the harmonic mean of precision and recall. The F1 score is calculated as follows:(3)$$\begin{array}{c} F1\;\mathrm{score}=\frac{2\times \mathrm{precision}\times \mathrm{recall}}{\mathrm{precision}+\mathrm{recall}}. \end{array}$$

As the baseline, we also present the results of a one-step recognition method.

### 5.2. Results

Figure 7 shows the results of the one-step recognition method. The numbers in the figure were normalized in each row, and the same was applied to the following confusion matrices. The mean F1 score was 65.5%.

Figure 8 and Figure 9 show the confusion matrices of the first and second steps of the proposed method, respectively. As shown in Figure 8, position recognition accuracy was around 90% for each position. The mean F1 score was 89.4%. For the shear-force recognition, the recognition accuracy was around 70% for each direction, as shown in Figure 9. The mean F1 score was 69.4%.

Figure 10 shows the comparison of the one-step recognition and proposed methods. In position recognition, the F1 score was 89.2% for the one-step recognition method, while it was 89.4% for the proposed method, showing no significant improvement. In calculating the position recognition of the one-step recognition method, even if the recognition of the shear-force direction was different, it was correct if the position was correct (e.g., if the input was P1-a and classification result was P1-b, it was considered to be correct recognition). In the recognition of shear-force direction, the F1 score was 65.5% for the one-step recognition method, while it was 69.4% for the proposed method, showing an improvement of 3.9 points. This may have been because the number of classes in each classifier could be reduced by two-step recognition, leading to improved recognition performance.

Figure 11 shows the F1 score of shear-force direction at each position. The F1 scores at P1 (62.7%) and P4 (68.8%) were lower than those at P2 (72.9%) and P3 (73.9%). This may have been due to the placement of the microphone and speaker, i.e., P2 and P3 were located on a straight line connecting the microphones/speakers, while P1 and P4 were off the straight line, as shown in Figure 6. The signal intensity of the acoustic signal was higher on the microphone/speaker straight line, suggesting that a greater change in acoustic characteristics was obtained when the participant was operating on that line.

The recognition accuracy for shear-force direction ‘a’ (65.6%) was lower than that for shear-force directions ‘b’ (70.0%), ‘c’ (72.0%), and ‘d’ (71.0%). One possible reason for this is the participants’ dominant hands. Since six out of the seven participants were right-handed, we consider that the results represent right-handed characteristics. The movement of direction ‘a’ was more unstable than that in other directions, as the hand was rotated outward from the body. We summarize the F1 scores of shear-force direction by the dominant hand in Table 2. Although there was only one left-handed participant, we confirmed that the F1 score of direction ‘b’, which is the direction in which the hand was rotated outward from the body for the left-handed participant, was also the worst in the four directions.

Figure 12 shows the F1 score of each participant, confirming individual differences in recognition performance. The F1 scores of Participants A and D were higher than the average. This is because they had experienced the proposed method multiple times; thus, the input actions were stable. The F1 scores of Participants B, E, and G were lower than the average. This is because the input actions were unstable due to their first experience with the proposed method. We expect the recognition performance to improve as the user becomes familiar with the proposed method.

## 6. Discussion

### 6.1. Recognition Performance

As shown in Figure 10, with the one-step recognition method, the F1 scores of position and shear-force direction were 89.2 and 65.5%, respectively, while those of the proposed method were 89.4 and 69.4%, respectively. Therefore, the proposed method is effective in recognizing both the pressing position and shear-force direction. However, even with this method, the F1 score of the shear-force direction was not high enough for practical use. To improve this performance, we need to collect more data from more participants. As shown in Figure 12, the F1 scores of Participants B, E, and G were lower than those of the other participants. This is because the machine learning model was not sufficiently trained to apply to all participants. Since each participant had a differently shaped finger and different ways of input, some participants did not fit this model. It is necessary to collect various data from a larger number of participants in the future.

### 6.2. Assumed Applications

Although the recognition accuracy needs to be improved for practical use, the proposed method can be applied to interfaces that enable multiple inputs from the same position and to curved-surface touch panels by taking advantage of the viscoelasticity of the gel. For example, a smartphone with multiple inputs from the same position by placing gel on the display and an interface that can be used while holding the steering wheel of a car by wrapping the gel around the steering wheel are possible. We plan to implement such an input interface, and investigate its operability and usability.

### 6.3. Gel Properties

Although we used one type of gel, gels have multiple factors to consider, e.g., material, hardness, shape, and thickness. In our preliminary investigation, we compared the 3 and 5 mm thick gels, and found that the sound pressure level obtained from the 5 mm thick gel was greater than that from the 3 mm thick gel; thus, we selected the 5 mm thick gel in this paper. A detailed investigation of the thickness and recognition performance is an issue to be addressed in the future.

Other factors need to be investigated such as its deterioration and deformation with each use. As the gel characteristics change due to the deterioration of gel, the obtained frequency characteristics are expected to change, and the recognition accuracy would be adversely affected. In such a case, it is necessary to periodically update the machine learning model. A long-term investigation on gel degradation is for a future study.

## 7. Conclusions

We proposed a two-step recognition method for recognizing both pressing position and shear force on gel plates using active acoustic sensing. The evaluation using a user-independent model confirmed that four pressing positions were recognized with an F1 score of 89.4%, and four shear-force directions were recognized with an F1 score of 69.4% using the proposed method. For future work, we plan to improve its recognition accuracy, implement an assumed input interface, and investigate the operability and usability of the method.

## Figures and Tables

**Figure 1 sensors-22-09951-f001:**
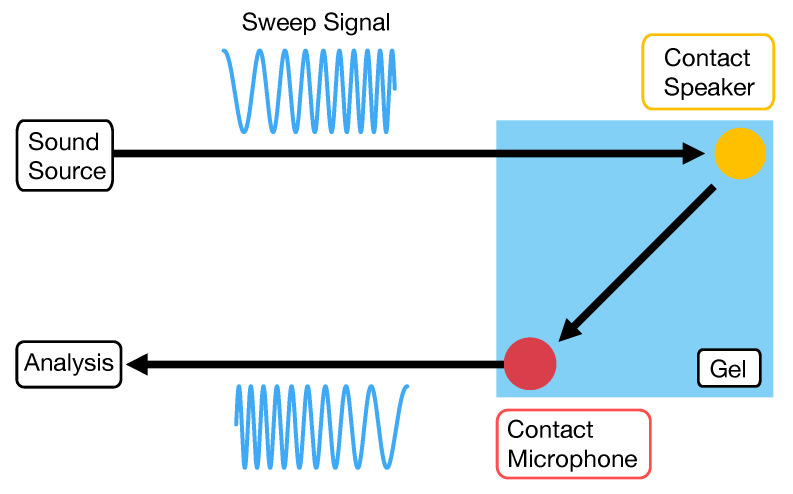
System configuration.

**Figure 2 sensors-22-09951-f002:**
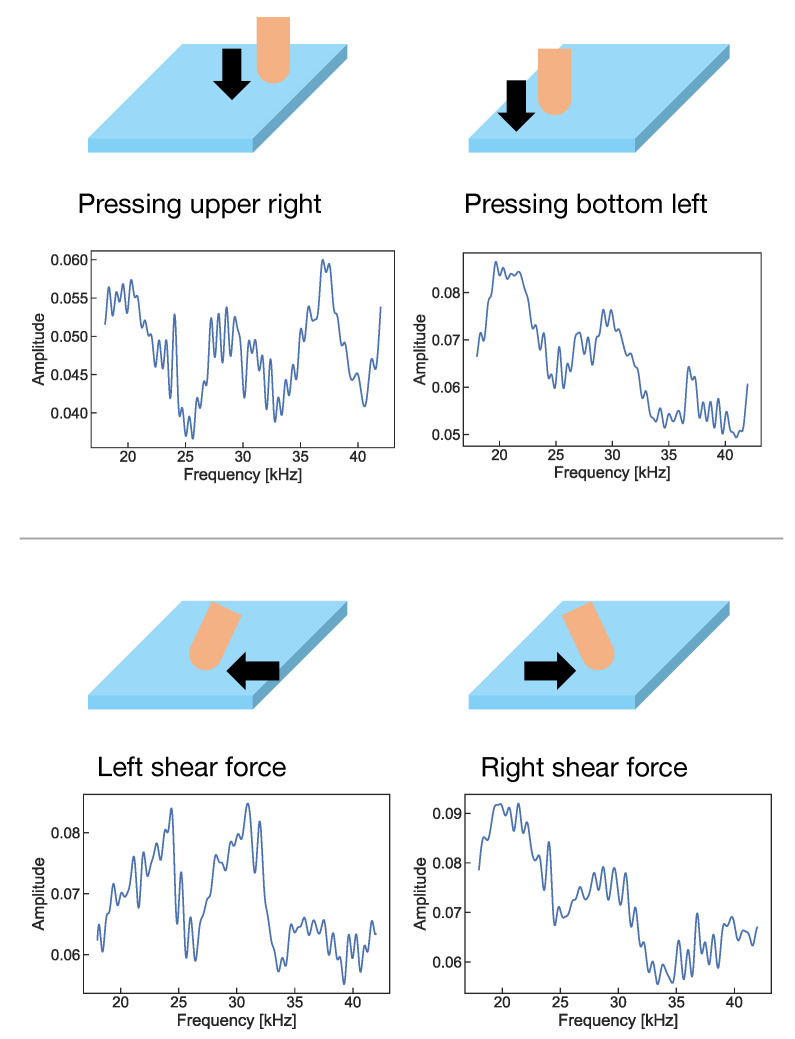
Change in frequency response.

**Figure 3 sensors-22-09951-f003:**
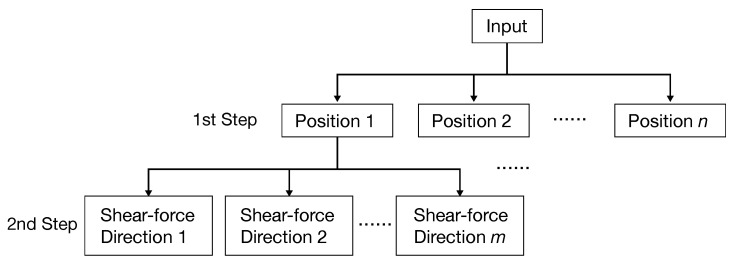
Overview of two-step recognition.

**Figure 4 sensors-22-09951-f004:**
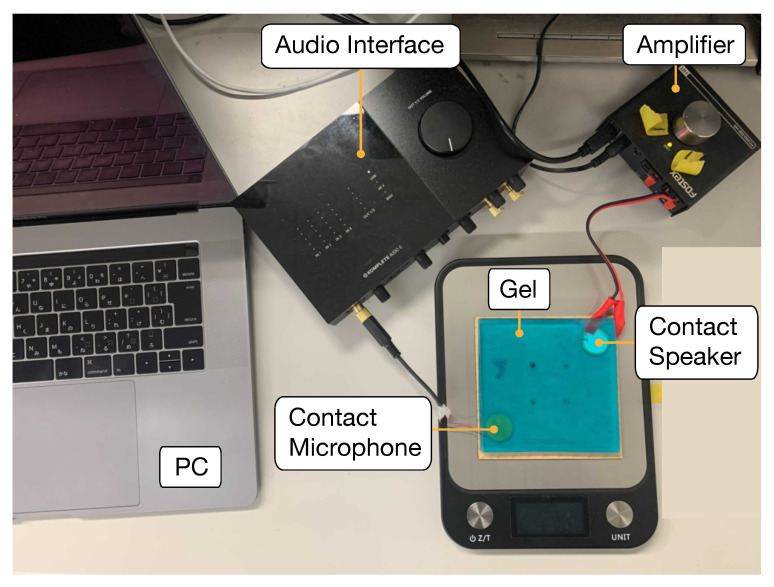
Implemented device.

**Figure 5 sensors-22-09951-f005:**
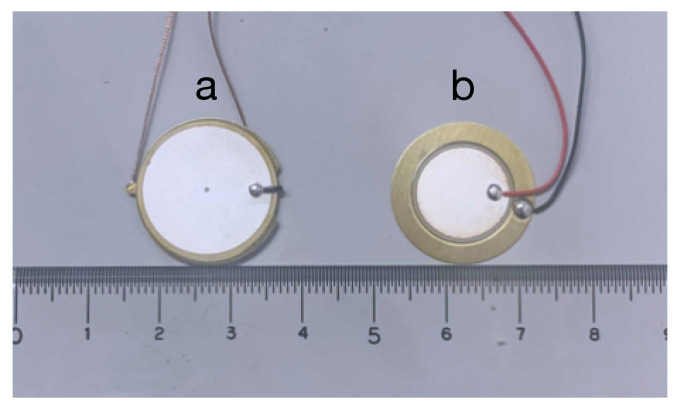
(**a**) Contact speaker and (**b**) contact microphone.

**Figure 6 sensors-22-09951-f006:**
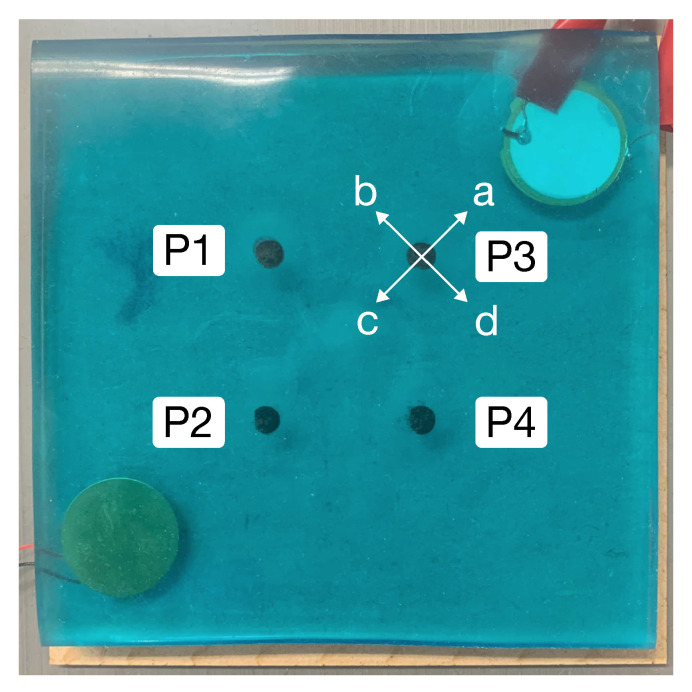
Pressing positions (P1–P4) and directions of shear-force directions (a–d).

**Figure 7 sensors-22-09951-f007:**
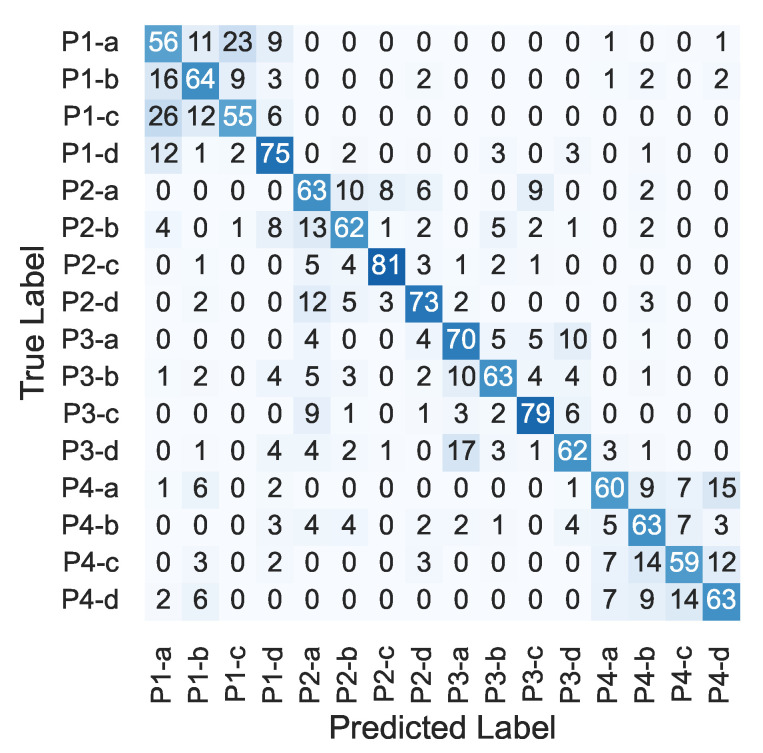
Confusion matrix of one-step recognition method [%].

**Figure 8 sensors-22-09951-f008:**
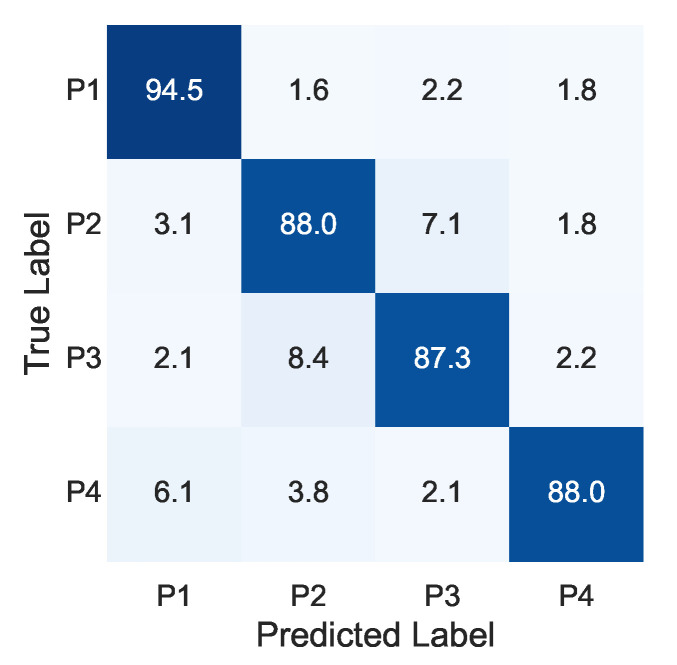
Confusion matrix of first step (%).

**Figure 9 sensors-22-09951-f009:**
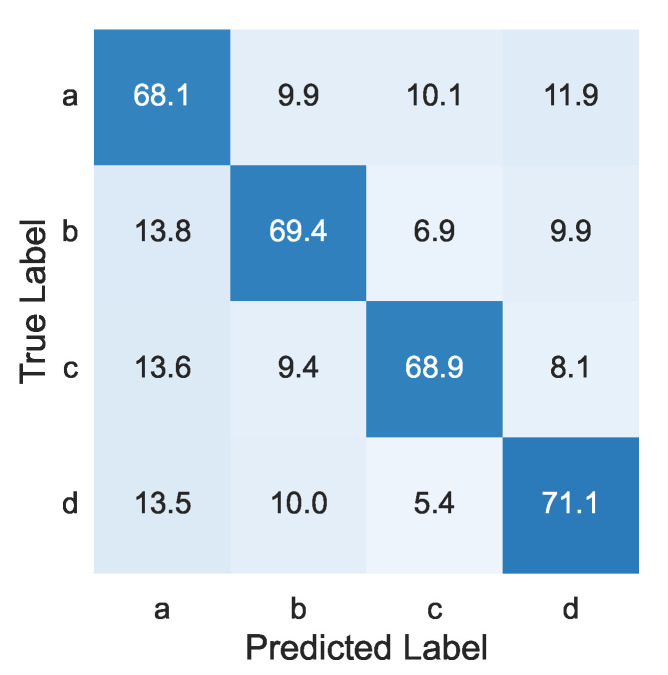
Confusion matrix of second step (%).

**Figure 10 sensors-22-09951-f010:**
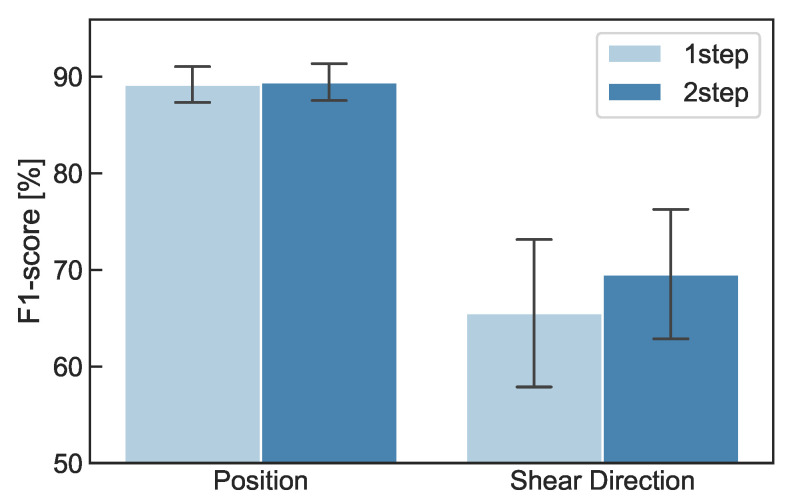
F1 score of one-step recognition and proposed methods (%).

**Figure 11 sensors-22-09951-f011:**
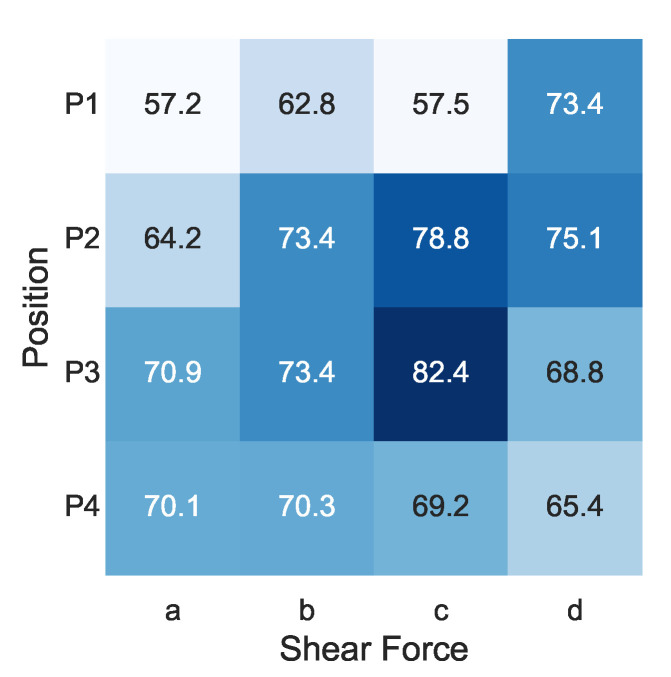
F1 score of shear force at each position (%).

**Figure 12 sensors-22-09951-f012:**
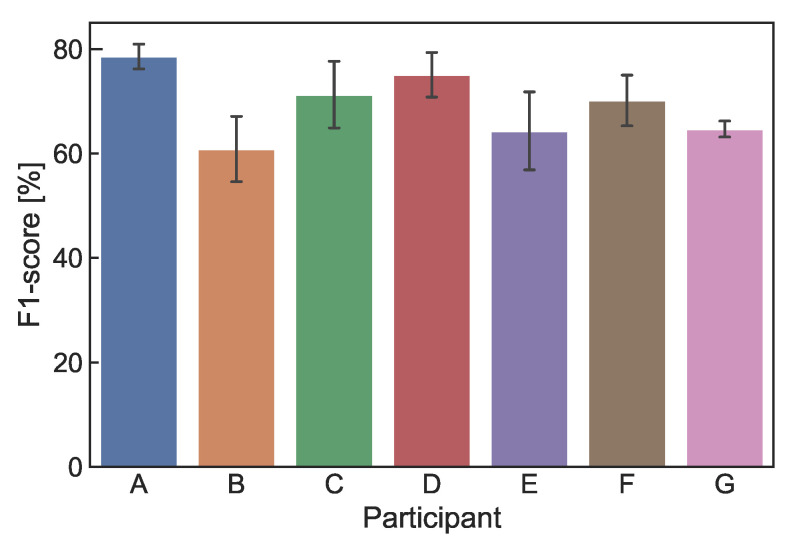
F1 score of each participant (%).

**Table 1 sensors-22-09951-t001:** Combinations of pressing pressure and shear force.

Pressure (g)	Shear Force (g)
300 g	300 g
300 g	600 g
600 g	300 g
600 g	600 g

**Table 2 sensors-22-09951-t002:** F1 scores of shear-force direction by dominant hand (%).

Shear-Force Direction	a	b	c	d
Left-handed	77.7	76.0	78.1	82.5
Right-handed	62.9	68.9	70.3	68.8

## Data Availability

The data presented in this study are available on request from the corresponding author. The data are not publicly available due to the fact that we have not received permission from the participants to publish. If you contact the corresponding author, we will contact the participant individually.

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
