# Peer review of "Method for Recognizing Pressing Position and Shear Force Using Active Acoustic Sensing on Gel Plates"

_sensors, 2022, doi:10.3390/s22249951_

Round 1

Author Response

Thank you for reviewing our paper. We submit the response to the comments. Please see the attachment. 

Reviewer 2 Report

The authors report two step recognition method for determining touch position and touch force using acoustic sensing. In this sensing, they transmit acoustic signals of various frequencies and analyze the emitted signal to elucidate the state of the sensing object. The proposed method utilizes simple set up consisting of gel plate and touch sensing module with acoustic wave generating speakers. Acoustic sensing is nothing new and several works published already on this field, however, the novelty of the work is sufficient due to the fact that the authors successfully managed to detect both position and sheer force using the construct. I recommend acceptance of the manuscript. The authors may consider the following minor points to improve their manuscript further:

1. the related work section or state-of-the-art is bit too long, I suggest a trimmed section here as some of the discussion goes to very basic and not totally relevant to a journal publication. 

2. The sampling frequency was chosen to be 96 kHz. Was it random? There was no justification given for that. 

3. Discussion on the gel , its physical and chemical properties are not presented. For a better understanding of the mechanism, the details of the gel must be provided. 

4. Also, the thickness of the gel layer was 5 mm, a better correlation could be achieved if the thickness could vary. 

5. I recommend a thorough English language check. Some sentences need rephrasing, e.g. the line 6-7. 

Author Response

(The authors gave the same response as above.)
